# Oral Function Rehabilitation with the Simplified Lauritzen Clinical Remount Technique in a Patient with Bimaxillary Alveolar Exostoses: A Case Report

**DOI:** 10.3390/healthcare10040682

**Published:** 2022-04-05

**Authors:** Chi-Hsiang Cheng, Ikiru Atsuta, Yuki Egashira, Kiyoshi Koyano, Yasunori Ayukawa

**Affiliations:** 1Section of Implant and Rehabilitative Dentistry, Division of Oral Rehabilitation, Faculty of Dental Science, Kyushu University, 3-1-1 Maidashi, Higashi-ku, Fukuoka 8128582, Japan; teikeishoudds@gmail.com (C.-H.C.); egashirayuki@dent.kyushu-u.ac.jp (Y.E.); koyano@dent.kyushu-u.ac.jp (K.K.); ayukawa@dent.kyushu-u.ac.jp (Y.A.); 2Division of Advanced Dental Devices and Therapeutics, Faculty of Dental Science, Kyushu University, 3-1-1 Maidashi, Higashi-ku, Fukuoka 8128582, Japan

**Keywords:** complete denture, malocclusion, exostoses, clinical remount

## Abstract

This case report describes a 70 year-old man with IVA lung cancer who required oral function rehabilitation by fabricating dentures with a simplified clinical remount technique. A pair of dentures were fabricated for a 70-year-old man with stage IVA lung cancer. Due to severe bimaxillary exostoses, the dentures could not properly extend and achieve a peripheral seal. The treatment philosophy was to stabilize the dentures and achieve proper function with optimized occlusion. The simplified Lauritzen clinical remount technique was performed at the time of denture delivery and 3 months later. After the second clinical remount procedure, the patient was able to eat meals with the dentures and maintained in a stable condition. Compared with the original technique, the simplified Lauritzen clinical remount omits the facebow transfer and keeps the condylar guidance setting and the Bennett angle unchanged during the adjustment. The prostheses are mounted to a type 3, non-arcon type articulator with anterior stop screws attached to the bilateral condylar parts. With the aid of anterior stop screws, the eccentric movement of dentures can be differentiated on a millimeter scale and balanced easily. It is effective to use occlusal-optimized dentures and the clinical remount technique, especially in difficult cases.

## 1. Introduction

The overall need for complete dentures is not likely to decrease in the near future [1]. However, edentulous patients are more likely to have impaired general health, which may result in reduced healing capacity, making it more challenging to achieve good condition of the denture-bearing tissues. Many edentulous patients have severely resorbed residual ridges and prominent anatomic landmarks or bony abnormalities [2].

Among various types of bony abnormalities, exostoses are considered an obstacle to successful denture construction [3]. Exostoses frequently hinder the attainment of a proper border seal of dentures, and the mucosa covering these hard convexities is thin and subjected to constant irritation during denture function [4] (p. 80). If preprosthetic surgery cannot be performed, the few available adjustments include relieving the tissue surfaces or shortening the flanges of the dentures. Denture bases without proper extension cannot be stabilized or distribute applied forces effectively. A denture without proper stability is a common cause of soreness and lack of retention [2].

If the tissue surface and the length of the flanges are limited, clinicians can only seek acceptable treatment results by adjusting the polished and occlusal surface of the dentures. A balanced occlusion ensures even pressure in all parts of the arch, which maintains the stability of the dentures and is more biomechanically favorable to the denture-bearing tissue and bone [5] (p. 273).

To achieve the desired occlusion, clinicians can either adjust the dentures in a patient’s mouth directly or on an articulator. Adjusting dentures on an articulator (known as a clinical remount) is by far the most accurate procedure to redefine occlusion [5] (p. 268). On the articulator, the dentures are firmly attached to the cast, rather than shifting on resilient tissues. The stone casts provide a stable and solid working environment. Thus, artificial teeth can be properly dried, and the occlusion errors can be easily identified and corrected [5] (p. 268).

Although the benefits of the clinical remount technique are well-described in multiple studies [6], only 1% of general practice dentists in the UK perform this technique frequently in daily practice. Most clinicians consider the clinical remount to be a time-consuming, complicated procedure and tend to instead directly adjust the occlusion of the dentures in the patient’s mouth [7].

In 1974, Lauritzen et al. [8] proposed an improved version of the clinical remount technique using an articulator with anterior stop screws attached to the condylar sphere housings. With these screws, the articulator can be set and locked at a certain eccentric position. The articulation of prostheses can then be differentiated on a millimeter scale and adjusted at each eccentric position. Multiple adjusted positions integrate smooth guiding paths for the prostheses, enabling the attainment of ideal occlusion and articulation. This system was further simplified in Japan for efficiency and easy manipulation [9,10].

Herein, we describe a case in which dentures were made to restore the impaired oral function of a patient with bimaxillary exostoses. Due to the exostoses, the peripheral seal of the dentures could not be achieved. Therefore, the treatment philosophy was to stabilize the dentures and achieve proper function with optimized occlusion via the simplified Lauritzen clinical remount procedure.

## 2. Case Report

### 2.1. General Information

A 70-year-old Japanese man was referred to Kyushu University Hospital for denture treatment. He had been diagnosed with stage IVA lung cancer, hypertension, asthma and chronic obstructive lung disease. He had also undergone several major surgeries, including femoral hernia repair at the age of 68 years and mediastinal lymph node dissection at the age of 69 years. No known drug allergies were reported. He stated that he had smoked a pack of cigarettes every day from the age of 20 until 69 years but had quit smoking thereafter.

Despite his compromised general health condition and continuing chemotherapy, the patient showed strong motivation to have further prosthodontic treatment. He stated that: “I cannot have decent meals, and would like to have new false teeth to aid”.

### 2.2. Intra-Oral Examination and Model Analysis

Intra-oral examination revealed maxillary total edentulism and mandibular partial edentulism with only the lower right second premolar remaining (Figure 1). The bony prominence around the bilateral tuberosity areas significantly reduced the space of the retrozygomatic fold. The bony prominence around the mandible extended from the lingual frenum to the retromylohyoid fossa bilaterally (Figure 2). Mild bony prominence was also present around the labial aspects of the maxilla and mandible, making it difficult to distinguish the peripheral frenula.

The morphology of both the maxillary and mandibular edentulous ridges hindered the proper extension of the denture bases. Although surgical removal of the exostoses was suggested, the patient rejected preprosthetic surgery due to personal preference. According to the classification system for complete edentulism, prosthodontic techniques of a specialized nature must be used to achieve an adequate result in such Class IV cases [11].

### 2.3. Clinical Remount Procedure

The upper complete denture and lower partial denture were fabricated using conventional methods. At the denture delivery appointment, the clinical remount technique was performed after the dentures were settled properly without pain.

*Centric relation* (CR) bite registration was taken with bite wax (Bitewax; GC Corporation; Tokyo, Japan). Instead of a facebow transfer, the dentures were mounted to the articulator (NDU-77; Takamiya, Saitama, Japan) with the mounting platform. This type 3, non-arcon type of articulator has anterior stop screws for the condylar sphere (Figure 3), enabling the use of the simplified Lauritzen technique. The Bennett angle and condylar guidance were set at average values of 7.5° and 30°, respectively, and were not altered during the whole process [9,10].

Selective grinding was then performed at the centric and eccentric positions [5] (p. 271). Balanced occlusion was first achieved at the CR. The dentures were then fixed at the edge-to-edge position of right eccentric movement (e.g., 3 mm from the CR position), and articulation at this position was balanced. The dentures were then set toward the CR position with a 1 mm advancement (e.g., 2 mm from the CR position) and the articulation was balanced again. The same procedure was repeated until the CR position was reached. The left and anterior eccentric movements were adjusted with the same method. After the *bilateral balanced occlusion* scheme was achieved, the dentures were removed from the articulator and polished.

When the patient received the dentures, he was instructed to try some sliced apple and peanuts [8,9,10] (p. 226). Pressure indicator paste (PIP) was applied at the intaglio surface while the patient consumed the food. The pattern of PIP was inspected and the intaglio surface was further relieved for preventing possible discomfort during masticatory function.

### 2.4. Treatment Course

The patient attended follow-up appointments every 1 to 2 weeks for the first 3 months. In the first 8 weeks, the flange and the intaglio surface of the dentures were carefully relieved as needed.

Approximately 3 months after denture delivery, the patient again reported difficulty chewing. The CR bite registration was taken and carefully verified, and the clinical remount technique was performed again. Before selective grinding, unilateral premature contact was noted. A balanced occlusal scheme was once again achieved (Figure 4), and the patient has since remained in a stable condition.

### 2.5. Treatment Results

After the treatment, the patient could eat soft food with the dentures. The pressure-sensitive sheet kit (Dental Prescale II, GC Corporation, Tokyo, Japan) indicated the increment of occlusal contact points (Figure 5).

Due to continuing chemotherapy, ulceration was occasionally noted around the mandibular exostoses. However, the patient stated that he could still consume soft meals using the dentures with tolerable discomfort. The treatment results were therefore acceptable despite the fragile underlying supporting tissues and the under-extended flange of bimaxillary dentures (Figure 6).

### 2.6. Further Treatment Plan

The treatment plan comprised monthly follow-up appointments. The clinical remount should be performed at least annually [8] (p. 227), [12] (p. 113), [13] to compensate for minor changes in the mandibular position, gradual changes of the temporomandibular joint and wearing of the artificial teeth [14]. If the reline or rebase procedure is performed, the clinical remount procedure should be performed [8,15,16] (p. 227). The occlusion and articulation in such difficult cases should be carefully managed so that the occlusal force during function is well distributed. This will enable the patient to experience the most comfort and ease possible in such limited conditions.

## 3. Discussion

To achieve the harmonious occlusion of complete dentures, the clinicians can adjust dentures at the chairside directly or with the remount procedure. Although it is also acceptable to adjust the occlusion of dentures directly in a patient’s mouth with articulating papers, the resiliency of the supporting tissue, moisture and inconsistent jaw movements create more spurious markings than the clinical remount [17]. Therefore, the premature contacts cannot be removed precisely, and the dentures will still shift and apply undistributed force to the underlying tissue during function, making it difficult to achieve ideal treatment results. Grinding spurious markings is also an unnecessary waste of the artificial teeth and will shorten the longevity of the dentures.

Most clinicians only consider the clinical remount technique a necessary procedure during the setting of new dentures. However, this technique is a useful way to refine the occlusion of processed dentures on an articulator and can be performed to reformat the occlusion of either newly made or intensively used dentures. Furthermore, the clinical remount technique not only enhances the accuracy of selective grinding procedures but also significantly reduces the chair-time and patient burden [10]. In the current aging society, these advantages provide the clinical remount technique with greater versatility and more applications, including refining the occlusion for bedridden patients or providing denture services in nursing homes or at voluntary dentistry events.

The simplified Lauritzen technique was introduced by a Japanese dentist named Hideo Kawahara in the 2010s [9]. Compared with the original technique proposed by Lauritzen et al. [8], the simplified technique omits the facebow transfer and requires CR bite registration only instead of multiple centric and eccentric records.

Unlike the original technique changes the settings of the condylar guidance and Bennett angle during the adjustment and between patients, the simplified technique utilizes mean values and keeps the settings unchanged. Instead of interpreting the dragging marks of eccentric movement and trying to achieve the desired occlusal scheme, it is easier to differentiate the mandibular eccentric movements on a millimeter scale and perform selective grinding at each setting. The omission of the facebow transfer, mounting the dentures with the CR bite only and setting the articulator at mean values all coincide with the recent trends of complete denture fabrication [18]. However, the difference in the total treatment efficiency between the original and the simplified technique remains unknown.

In the present case, the second clinical remount adjustment was performed 3 months after denture delivery. Multiple factors may cause mild occlusal discrepancies, including the adjustment of the flange and intaglio surface of the dentures, water resorption and distortion of the denture bases, settling of the denture-bearing tissue and neuromuscular adjustment of the masticatory system [8,13,14] (p. 159). Therefore, when the patient presented with occlusal-related complaints, the clinical remount was repeated without hesitation.

As described above, the stability of the dentures in the present case is solely reliant on the occlusion. Although the lower right second premolar remained and provided a certain amount of stability for the lower prosthesis, it would only serve as a fulcrum point of a shifting denture if the occlusion was not properly managed. In such a circumstance, even if the intaglio surfaces of the prostheses were properly relieved, discomfort would persist and the remaining abutment teeth would be overloaded with lateral forces. Efforts should be made to establish well-distributed occlusal contact points and interference-free eccentric pathways for jaw movements. Therefore, the clinical remount was repeated to achieve a balanced occlusion that was as precise as possible.

Despite the scarcity of evidence-based guidelines for the clinical remount procedure for delivered prostheses [9,19], the value of this technique and the ideal occlusal scheme of dentures should not be underestimated, especially in difficult cases. Clinicians should understand this technique and utilize it when appropriate.

## 4. Conclusions

In the present case, the chief complaint was resolved, and the patient was able to consume meals with the dentures. In difficult cases, the value of occlusion-optimized dentures should not be underestimated. The clinical remount technique should be considered solely as an accurate method to achieve the desired occlusion. This philosophy will broaden the potential applications of the clinical remount technique. The simplified Lauritzen technique differentiates the eccentric movements of mounted prostheses and makes the clinical remount procedure easier. Despite further evaluation of this technique being required, the value of this technique should not be overlooked.

## Figures and Tables

**Figure 1 healthcare-10-00682-f001:**
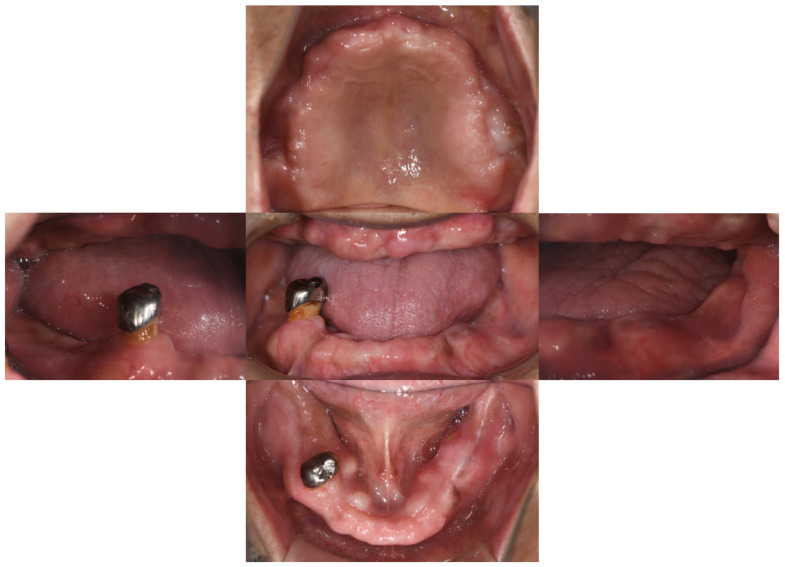
Intra-oral examination. The maxilla was completely edentulous. The mandible had only the lower right second premolar remaining.

**Figure 2 healthcare-10-00682-f002:**
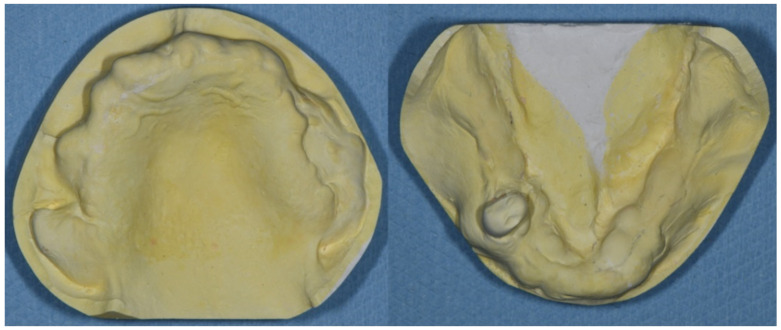
(**Left**). The maxillary study model. The bony exostoses around bilateral tuberosity areas hindered the proper extension and the peripheral seal of the maxillary denture. (**Right**) The mandibular study model. The lingual bony exostoses extended from the lingual frenum to the bilateral retromylohyoid fossa area.

**Figure 3 healthcare-10-00682-f003:**
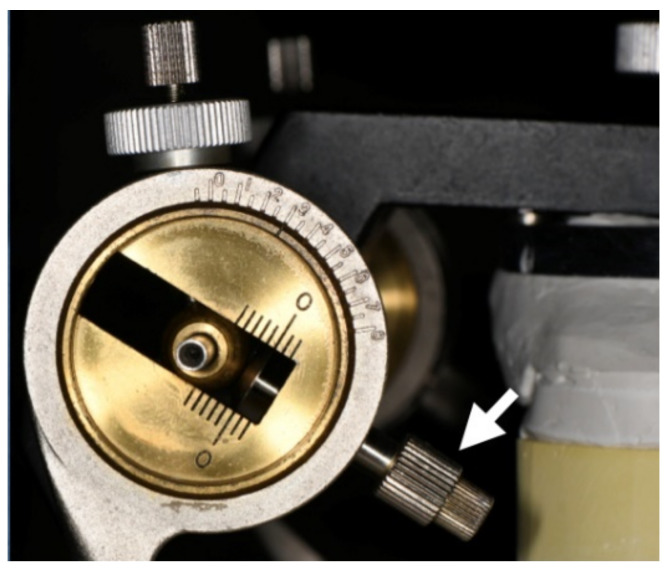
Please note the condylar parts of the articulator with anterior stop screws (white arrow). The articulator can be set on specific eccentric positions with the aid of these screws, making the clinical remount procedure easier to perform.

**Figure 4 healthcare-10-00682-f004:**
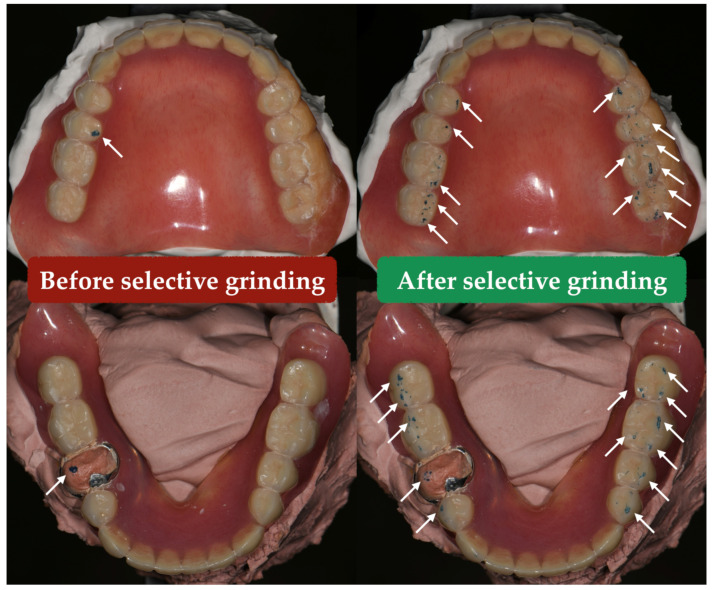
At the *centric relation* position, there was premature contact (white arrow) before selective grinding. After adjustment, the balanced occlusal scheme was achieved (white arrow).

**Figure 5 healthcare-10-00682-f005:**
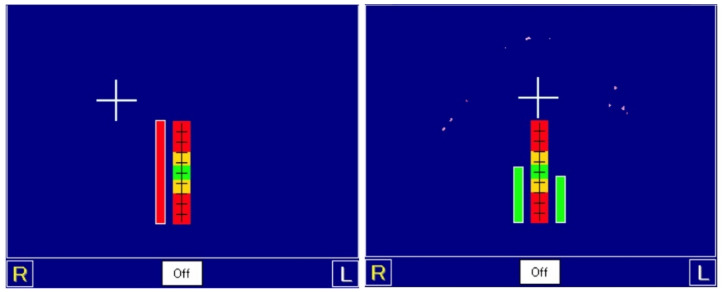
Pressure-sensitive sheet showing the occlusal contact points at the *maximal cuspal position*. The left image shows the contact point before the adjustment. The right image shows the increase in the contact points after selective grinding.

**Figure 6 healthcare-10-00682-f006:**
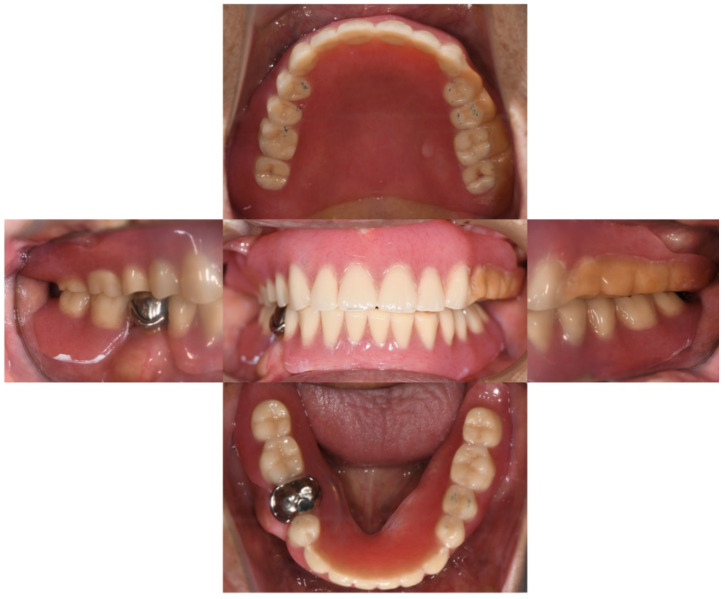
The intra-oral photo of the delivered prostheses. The shortened denture flange was obvious around the left tuberosity area.

## Data Availability

Not applicable.

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
