# Peer review of "Oral Function Rehabilitation with the Simplified Lauritzen Clinical Remount Technique in a Patient with Bimaxillary Alveolar Exostoses: A Case Report"

_healthcare, 2022, doi:10.3390/healthcare10040682_

Round 1

Reviewer 1 Report

First of all, please allow me to congratulate the authors for  attempting to undertake this study which I found quite interesting and useful as a reference for further study in Oral and Dental Diseases. The manuscript itself is well-written and well-structured and I have to also  commend the author for this matter. However, I may require some clarification on the following issues. The paper is well organized and easy to follow. To improve the readability, it is recommended that the text is checked by a native english speaking person as many of sentences might be misunderstood. I suggest a revision of the grammar structure by an expert editor in revising manuscripts.

Author Response

The modification report for Reviewer #1

Dear Reviewer #1:

Thank you for pointing out the flaws of our article. We had modified the manuscript according to your comment. These details included:

  1. Line 125: “The intaglio surface was further adjusted as needed.”
    Explain what should be in the intaglio surface adjusted as needed?

> Response: Thank you for your comments. We have modified the manuscript to:

“The pattern of PIP was inspected and the intaglio surface was further relieved for preventing possible discomfort during masticatory function.”

  1. Line 148: “Despite multiple negative factors.”
    Explain multiple negative factors.

> Response: We have modified the article and attached intra-oral photos of the post-treatment condition.

“The treatment results were therefore acceptable, despite the fragile underlying supporting tissues and the under-extended flange of bimaxillary dentures (Figure 6).”

  1. Line 217: “In an aging society, this technique…”

Please use another term.

> Response: The conclusion part was rearranged into:

“In the present case, the chief complaint was resolved and the patient was able to consume meals with the dentures. In difficult cases, the value of occlusion-optimized dentures should not be underestimated. The clinical remount technique should be considered solely as an accurate method to achieve the desired occlusion. This philosophy will broaden the potential applications of the clinical remount technique. The simplified Lauritzen technique differentiates the eccentric movements of mounted prostheses and makes the clinical remount procedure easier. Despite the further evaluation of this technique is required, the value of this technique should not be overlooked.”

Once again, we appreciate your patient reading and delicate check for our manuscript.

Sincerely,

Chi-Hsiang Cheng & Ikiru Atsuta

Reviewer 2 Report

The article is well written , but needs some changes. 

Author Response

The modification report for Reviewer #2

Dear Reviewer #2:

Thank you for pointing out the flaws of our article. We had modified the manuscript according to your comment. These details including:

Abstract:

  1. Remodify the sentence from 14 -15. This case report describes a 70 year old man with IVA lung cancer

who required oral function rehabilitation by fabricating dentures with simplified clinical remount technique.

> Response: Thank you for the advice. Modified the sentence from Line 14 - 15.
“This case report describes a 70-year-old man with IVA lung cancer who required oral function rehabilitation by fabricating dentures.”

Introduction:

  1. Split the sentence from Line 41 – 43.

> Response: The modified sentences are as followed:
“Denture bases without proper extension cannot be stabilized or distribute applied forces effectively. A denture without proper stability is a common cause of soreness and lack of retention.”

  1. Color change from Line 62.

> Response: Thank you for pointing out. This mistake was already corrected.

Case Report:

  1. Line 78 – 79: the post-surgical history is explained nicely. but follow it according to the year of appearance (first explain what happened at 68 years and then proceed to the next)

> Response: Thank you for your comments. Modified the sentence from Line 78 - 79.
“He had also undergone several major surgeries, including femoral hernia repair at the age of 68 years and mediastinal lymph node dissection at the age of 69 years.”

  1. Line 87-90: Clinical photographs will give better picture with the cast. Its better to add clinical pictures during intraoral examination and as well as post pic after delivering prosthesis/ dentures.

> Response: The intraoral photos before and after prosthodontics treatments were added.
Thank you for your advice. The manuscript looks more professional now.

Discussion:

  1. Discussion needs to be rearranged. The discussion starts abruptly. I recommend starting the discussion

by discussing about clinical remount technique and then simplified Lauritzen technique. Later, you can discuss about your case and the importance of the steps whenever required.

> Response: Response: Thank you for the advice. The discussion part was heavily modified according to your comment. We totally agreed that the discussion part before modification was a bit blurry. After modification, the discussion starts with the clinical remount procedure then the simplified Lauritzen technique. Some discussion of the current case is placed in the last part of the discussion. Thank you again.

Once again, we appreciate your patient reading and delicate check for our manuscript.

Sincerely,

Chi-Hsiang Cheng & Ikiru Atsuta

Author Response

The modification report for Reviewer #3

Dear Reviewer #3:

Thank you for pointing out the flaws of our article. We had modified the manuscript according to your comment. These details including:

  1. The simplified Lauritzen technique maked the clinical remount procedure easier and faster. However, one description is insufficient to draw conclusions.

> Response: The conclusion part was re-arranged. We point out the treatment results of the current case and the importance of harmonious occlusion. Then the simplified Laurtizen technique was described as an “easier” method instead of the “easier and faster” one. The modified conclusion part is followed:

“In the present case, the chief complaint was resolved and the patient was able to consume meals with the dentures. In difficult cases, the value of occlusion-optimized dentures should not be underestimated. The clinical remount technique should be considered solely as an accurate method to achieve the desired occlusion. This philosophy will broaden the potential applications of the clinical remount technique. The simplified Lauritzen technique differentiates the eccentric movements of mounted prostheses and makes the clinical remount procedure easier. Despite the further evaluation of this technique is required, the value of this technique should not be overlooked.”

  1. I propose to withdraw the references from 1971,1974,1986, as there are many manuscripts on this topic from our century.

> Response: Thank you for your comments. The cited references were modified. The original research from Murphy et al (1971) was removed. The citations from Syllabus of Complete Denture 4th edition (1986) were replaced with other mainstreamed textbooks and articles.

However, we sincerely suggest remaining Atlas of Occlusal Analysis written by
Dr. Lauritzen (1974), because the simplified clinical remount technique demonstrated in the current manuscript was originated from this book. Dr. Kawahara (2016) also mentioned the technique he modified was originated from Dr. Lauritzen’s textbook.
Therefore, multiple references were cited at the same time while citing the textbook from Dr. Lauritzen. to enhance the persuasion of our manuscript.

Once again, we appreciate your patient reading and delicate check for our manuscript.

Sincerely,

Chi-Hsiang Cheng & Ikiru Atsuta
